# Investigation of Melioidosis Outbreak in Pig Farms in Southern Thailand

**DOI:** 10.3390/vetsci7010009

**Published:** 2020-01-14

**Authors:** Wiyada Kwanhian, Treenate Jiranantasak, Aleeza T. Kessler, Bryn E. Tolchinsky, Sarah Parker, Jirarat Songsri, Suebtrakool Wisessombat, Kawinsaya Pukanha, Vincentius A. Testamenti, Pacharapong Khrongsee, Somporn Sretrirutchai, Jedsada Kaewrakmuk, Jitbanjong Tangpong, Apichai Tuanyok

**Affiliations:** 1Center of Excellence in Research for Melioidosis, Department of Medical Technology, School of Allied Health Sciences, Walailak University, Nakhon Si Thammarat 80161, Thailand; kwiyada@wu.ac.th (W.K.); jirarat.so@wu.ac.th (J.S.); sueptrakool.wi@wu.ac.th (S.W.); samita.pukunhaa@gmail.com (K.P.); rjitbanj@wu.ac.th (J.T.); 2Department of Infectious Diseases and Immunology, College of Veterinary Medicine, and the Emerging Pathogens Institute, University of Florida, Gainesville, FL 32608, USA; treenate@ufl.edu (T.J.); aleezatkessler@ufl.edu (A.T.K.); brynemilee@ufl.edu (B.E.T.); sparkn1902@gmail.com (S.P.); vincent.arca@gmail.com (V.A.T.); firstpachar@ufl.edu (P.K.); 3Faculty of Medicine, Prince of Songkla University, Hatyai 90110, Thailand; Sompornsre@gmail.com; 4Faculty of Medical Technology, Prince of Songkla University, Hatyai 90110, Thailand; jedsada.k@psu.ac.th

**Keywords:** swine melioidosis, *Burkholderia pseudomallei*, Thailand, pig, abattoir

## Abstract

Melioidosis, caused by the Gram-negative bacterium *Burkholderia pseudomallei*, is a potentially life-threatening infection that can affect humans and a wide variety of animals in the tropics. In December 2017, a swine melioidosis case was discovered during a meat inspection at a privately-owned slaughterhouse in Nakhon Si Thammarat Province in southern Thailand. The infection, which continued for several months, caused a dispute about where the disease began. An environmental investigation into two farms—both involved in raising the first infected pig—ensued. Through genetic analysis, the investigation revealed that a contaminated water supply at one farm was the probable source of infection. The three local sequence types identified in the investigation were types 51, 298 and 392.

## 1. Swine Melioidosis

Melioidosis, an infection endemic to Southeast Asia, northern Australia, and other tropical areas, is caused by the soil-dwelling Gram-negative bacterium *Burkholderia pseudomallei*, which is known to contaminate water sources [1]. The infections are cyclic and correlate with rainy seasons, which may be due to environmental disruptions such as flooding. While creating a major health concern for humans, *B. pseudomallei* is also a threat to various animals including reptiles, marine mammals, and domestic animals [2]. In Thailand, a more recent report has revealed that infection by the bacterium commonly occurs in farm animals, with goats having the highest incidence rate, followed by pigs and cattle [3]. Choy and colleagues reported three possible zoonotic cases in Australia including a butcher who directly handled infected meat or animals [2]. Although information about the zoonotic potential of *B. pseudomallei* is very limited, we believe that *B. pseudomallei* can be transmitted to humans through the butchering process or by the consumption of infected meat. Therefore, the infection of agricultural livestock could represent a public health hazard.

In December 2017, a group of 20 five-month-old pigs being finished for market weight were received at a private abattoir. One pig in the group was discovered to have a liver infection, later confirmed as melioidosis by bacterial culture. A dispute regarding the location of the exposure then arose between the two farms where the pig had previously spent time. The infected pig lived on two separate farms located in Nakhon Si Thammarat Province, referred to as Farm A and Farm B. The pig was born and raised for about four weeks in Farm A, then transferred 10 km away to Farm B and resided there for four months before the slaughterhouse purchased it. The pig had a normal appetite and showed no clinical signs upon arrival at the lairage for ante-mortem inspection. The infection was identified as a result of two large abscesses in the liver observed during meat inspection (Figure 1a). The organ was sent to our laboratory at Prince of Songkla University Animal Hospital for diagnosis. Since melioidosis was suspected, the abscess was cultured by a standard technique using MacConkey agar and streaked onto two extra plates of Ashdown’s agar, a selective medium for *B. pseudomallei*. Numerous bacterial colonies were observed (Figure 1b) and further identified as *B. pseudomallei* by biochemical tests, the latex agglutination test, and TTS1 real-time PCR [4].

## 2. Environmental Source of the Infection and Additional Cases

The investigation began with examining for the presence of *B. pseudomallei* in soil and water at both farms. Both used underground water sources from unchlorinated, single pump-per-farm systems for feeding and cleaning. Soil samples from both farms were cultured using the consensus guidelines previously developed [5]. Three 250 mL water samples were collected from a main pipe at each farm. Each sample was filtered through a 0.45 µm membrane using the Whatman MBS I Microbiological Membrane Filtration System (Rocker Scientific, New Taipei City, Taiwan). Each membrane was then placed onto an Ashdown’s agar plate and incubated at 37 °C for 48 h. We found that all water samples from Farm B grew many bacterial colonies (Figure 1c), while the samples from Farm A did not. Thirty-six suspected *B. pseudomallei* colonies were collected from all three agar plates for further species confirmation by TTS1 real-time PCR [4]. All 36 colonies were positive for *B. pseudomallei* through this assay. Multi-locus sequence typing (MLST) was then used to identify the sequence type (ST) of these *B. pseudomallei* isolates as previously described [6]. MLST analysis showed that all 36 *B. pseudomallei* isolates belonged to a single sequence type (ST392), which was the same as the isolates from the infected pig identified in December 2017 (Table 1). In addition, one of the 17 soil samples collected from each farm grew *B. pseudomallei* (Figure 1d). Five *B. pseudomallei* isolates chosen from each positive soil sample were further genotyped by MLST. All soil isolates from Farm A had a novel ST, designated as ST1721, while the isolates from Farm B were the same ST (ST392), as had been found in the infected pig and water samples. Furthermore, five months after the incident, the same abattoir submitted six additional abscess specimens from the lungs, liver, and spleen of pigs raised in multiple farms to our laboratory for analysis. Out of six samples, five grew *B. pseudomallei* (Table 1). MLST analysis showed that these infections were caused by strains with ST392 and two other STs: 51 and 298. Based on the current public MLST database (https://pubmlst.org/bpseudomallei/), all three of these STs have previously been reported from Thailand, with ST392 and ST51 reported to have been discovered in water samples collected in southern Thailand around 50 years ago [7,8]. This suggests that these porcine infections were caused by local genotypes prevalent in southern Thailand.

## 3. Serological Surveys

We noted that many pig farms in this province purchased piglets from Farm A where we had isolated *B. pseudomallei* from the soil. A serological survey of antibodies to *B. pseudomallei* was then conducted in sows and piglets bred at this farm. Nineteen pairs of serum samples from each sow (2–3 years old) and its piglet (four weeks old) were collected for serological analysis by the indirect hemagglutination (IHA) test. IHA test results showed that all piglets were serological negative with titer ≤1:20, while most sows had titers ≤1:80, except for a single sow with a titer of 1:320. This finding could suggest that this seropositive sow may have been exposed to *B. pseudomallei* through environmental contamination in this farming system. However, no further investigation was conducted to determine whether this pig had an ongoing infection.

## 4. Discussion

Sprague and Neubauer reported in 2004 that the major symptoms of swine melioidosis tended to be subclinical. The lungs could be infected via inhalation of the bacterium and were the most affected organ within pigs. Lung infection can present with nodular lesions or in clusters of pneumonic areas. Additional organs that might form nodular lesions are the spleen, liver, kidneys, lymph nodes, and skin, though these are less common. Others rarely seen, but clinically significant, symptoms include fever, diarrhea, coughing, dyspnea, discharge from eyes and nose, emaciation, and uncoordinated movement [9]. In 1962, a report from Malaysia described that melioidosis was acute more frequently in young pigs than in adult pigs, which often presented with chronic infection [10]. However, a different finding was discovered when the experimental *B. pseudomallei* infection was conducted in pigs. This study revealed that *B. pseudomallei* caused a chronic infection even after the intravenous challenge of two-month-old pigs with a dose of 5.0 × 10^9^ CFU [11]. There have been several documented swine melioidosis outbreaks in eastern and northern Thailand both within and outside endemic areas [12]. In Australia, a melioidosis outbreak was also reported in an intensive piggery 40 km north of Townsville, Queensland during 1979–1980. In that outbreak, *B. pseudomallei* was isolated from both soil and water, and the source of infection was linked to the unfiltered-unchlorinated water supply [13].

## 5. Conclusions

The present study indicates that pig farmers in southern Thailand should be aware of melioidosis stemming from water systems contaminated with *B. pseudomallei*. This investigation suggests that farmers should consider taking measures to disinfect their water in order to keep their livestock healthy and their customers safe. Melioidosis in pigs is not currently a part of the animal disease control program in Thailand. To prevent a further outbreak, Farm B agreed to temporarily shut down and was instructed to implement a chlorination system for its water.

## Figures and Tables

**Figure 1 vetsci-07-00009-f001:**
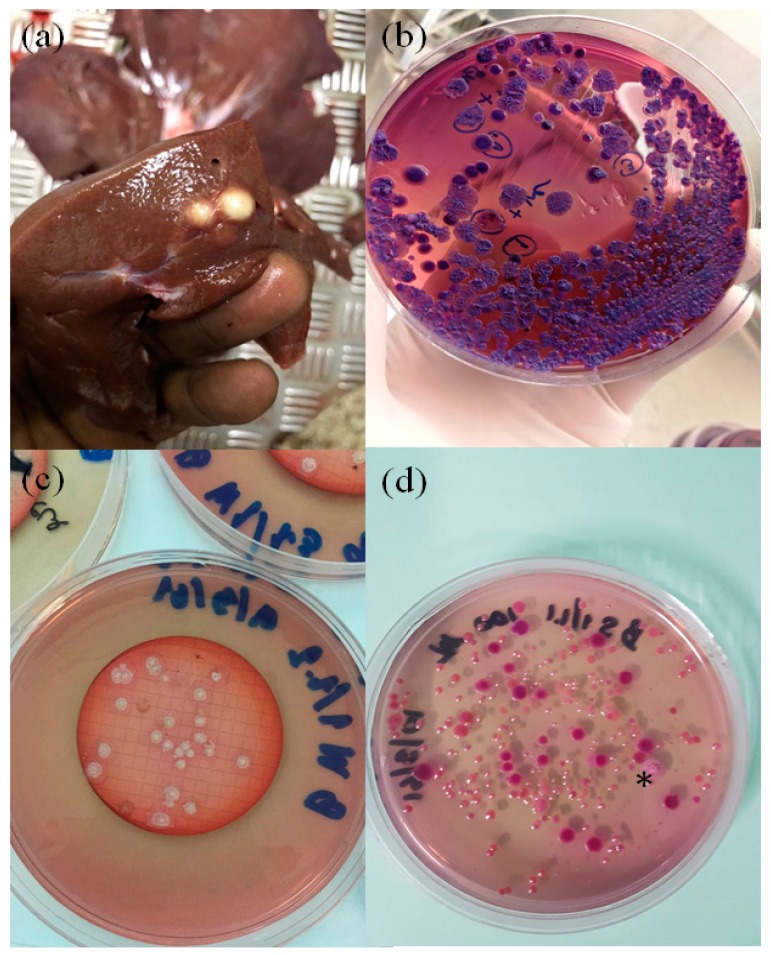
Melioidosis in a pig farm linked to *B. pseudomallei* contamination in the soil and water supply. Two large liver abscesses from a pig were discovered by an unprotected abattoir worker in December 2017 (**a**), and *B. pseudomallei* was isolated in pure culture on Ashdown’s agar (**b**). Water and soil collected from this farm cultured *B. pseudomallei* (**c**,**d**, respectively) possessing the same sequence type 392 as that found in the infected pig. Note: *, a confirmed *B. pseudomallei* colony grown from soil (**d**).

**Table 1 vetsci-07-00009-t001:** The incidences and the environmental investigation.

Date of Incidence	Type of Specimens	No. of Culture Positive Samples/Total	Farm	ST	No. of *B. pseudomallei* Isolates Analyzed by MLST
12/20/17	Liver abscess	2/2 *	Farm B	392	5
3/15/18	Soil	1/17	Farm A	1721	5
3/15/18	Soil	1/17	Farm B	392	5
3/15/18	Water	0/3	Farm A	N/A	N/A
3/15/18	Water	3/3	Farm B	392	36
05/22/18	Liver abscess	0/5 *	Not specified	N/A	N/A
05/26/18	Lung abscess	5/5 *	Not specified	51	5
06/05/18	Spleen abscess	5/5 *	Farm B	392	5
06/05/18	Liver abscess	5/5 *	Farm B	392	5
06/15/18	Liver abscess	5/5 *	Not specified	298	5
06/23/18	Lung abscess	5/5 *	Not specified	51	5

* is the number of abscesses from each infected organ tested; N/A, not applicable as negative for *B. pseudomallei*.

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
