# Peer review of "Investigation of Melioidosis Outbreak in Pig Farms in Southern Thailand"

_vetsci, 2020, doi:10.3390/vetsci7010009_

Round 1

Reviewer 1 Report

In the manuscript by Kwanhian et al., the authors report a melioidosis outbreak in pig farms in Thailand. The authors nicely describe the importance of reporting the swine melioidosis and describe their observations clearly using multiple methods of testing for presence of B. pseudomallei. I have no major issues with manuscript, but I think a discussion of a more recent report on animal melioidosis can be added (i.e. Limmathurotsakul et al 2012 Emerging Infect Dis). Additionally, the manuscript needs to be carefully revised for English grammar to improve the manuscript.

Minor corrections:

-Line 37-38: A more recent report on animal melioidosis can be added here.

-Line 44: Instead of “infection withi its liver”, “liver infection” can be used for clarity.

- Line 45: “organ culture” should be “bacterial culture”

-Line 48: “raised about 4 weeks” should be “raised for about 4 weeks”

-Line 74-76: The sentence should be revised. It can be edited as “…submitted 6 additional abscess specimens from lungs, livers and spleens of pigs raised in multiple farms to our laboratory for analysis. Out of 6 samples, 5 grew…”

Line 110-111: The sentence should be revised. It can be edited as “… melioidosis was acute more frequently in young pigs than in adult pigs, which present often with chronic infection.”

-Line 113-114: The sentence should be revised. It can be edited as “… a chronic infection even after intravenous challenge of two-month-old pigs with a high dose of… .”

-Line 115-116: The sentence should be revised. It can be edited as “… both within and outside endemic areas.”

-Lines 116 and 120: please re-format the citations.

-Line 122-123: The sentence should be revised. It can be edited as “… keep their livestock healthy and their customers safe.”

-Line 125: “temporary” shut down? Otherwise, why would they agree to install a chlorination system?

Author Response

Reviewer#1

Comments and Suggestions for Authors

In the manuscript by Kwanhian et al., the authors report a melioidosis outbreak in pig farms in Thailand. The authors nicely describe the importance of reporting the swine melioidosis and describe their observations clearly using multiple methods of testing for presence of B. pseudomallei. I have no major issues with manuscript, but I think a discussion of a more recent report on animal melioidosis can be added (i.e. Limmathurotsakul et al 2012 Emerging Infect Dis). Additionally, the manuscript needs to be carefully revised for English grammar to improve the manuscript.

Authors’ response: Thank you for the reviewer’s comments. We have modified the manuscript to reflect the comments and suggestions. The suggested reference “Limmathurotsakul et al. 2012” was not cited in the original version by error. In the revised version this article was added as a reference no. 3.

Minor corrections:

-Line 37-38: A more recent report on animal melioidosis can be added here.

Authors’ response: The statement is now written “In Thailand, a more recent report has revealed that infection by the bacterium commonly occurs in farm animals, with goats having the highest incidence rate, followed by pigs and cattle [3].

-Line 44: Instead of “infection withi its liver”, “liver infection” can be used for clarity.

Authors’ response: This change has been made in the revised manuscript.

- Line 45: “organ culture” should be “bacterial culture”

Authors’ response: This change has been made in the revised manuscript.

-Line 48: “raised about 4 weeks” should be “raised for about 4 weeks”

Authors’ response: This change has been made in the revised manuscript.

-Line 74-76: The sentence should be revised. It can be edited as “…submitted 6 additional abscess specimens from lungs, livers and spleens of pigs raised in multiple farms to our laboratory for analysis. Out of 6 samples, 5 grew…”

Authors’ response: This change has been made in the revised manuscript.

Line 110-111: The sentence should be revised. It can be edited as “… melioidosis was acute more frequently in young pigs than in adult pigs, which present often with chronic infection.”

Authors’ response: This change has been made in the revised manuscript.

-Line 113-114: The sentence should be revised. It can be edited as “… a chronic infection even after intravenous challenge of two-month-old pigs with a high dose of… .”

Authors’ response: This change has been made in the revised manuscript.

-Line 115-116: The sentence should be revised. It can be edited as “… both within and outside endemic areas.”

Authors’ response: This change has been made in the revised manuscript.

-Lines 116 and 120: please re-format the citations.

Authors’ response: Both references have been reformatted in the revised manuscript.

-Line 122-123: The sentence should be revised. It can be edited as “… keep their livestock healthy and their customers safe.”

Authors’ response: This change has been made in the revised manuscript.

-Line 125: “temporary” shut down? Otherwise, why would they agree to install a chlorination system?

Authors’ response: Yes, the shutdown was temporary until the chlorination system was installed. This was forced by the contract agreement Farm B had with the slaughterhouse.

Reviewer 2 Report

This paper describes investigations into porcine melioidosis in sothern Thailand following the identification of an initial case in 2017.  A range of investigations (molecular epidemiology, environmental sampling, serology) was undertaken and a posible link to a water supply in one farm was made on the basis of these.

The manuscript would benefit from careful review by one of the authors whose first language is English.  Tenses in particular require attention and I have not attempted to list every instance where these need to be changed.  I have mentioned a few examples below where changes to the English would improve the paper.

My main comment is that Table 1 would benefit from the addition of some numbers (samples taken, samples positive, colonies tested by MLST etc.). 

It would also be useful if some more detailed information could be provided about the animals that were culture positive in 2018.  For example, did these have any links with Farms A and B, were the 2 ST 392 isolates from the same animal?

Since it is now 2 years after the initial case, it would also be useful to have some follow up on Farm B. Has it reopened, what happened about the water supply, have there been any more cases from there etc.?

Minor suggestions to improve the paper:

Line 29. Change 'infection' to 'investigation'. Line 33. Melioidosis is also endemic in other tropical areas. Line 37. Change 'also is' to 'is also'. Line 39. Reference 3 is not really relevant to this statement - Limmaturotsakul et al. Melioidosis in animals, Thailand, 2006-2010. Emerg Infect Dis, 2012 would be more appropriate. Line 40. Change 'zoonotic potentials' to 'the zoonotic potential'. Lines 40-2. In fact there is very little hard evidence of zoonotic transmission of melioidosis and this section is purely hypothetical.  If these statements are to be included then they should be referenced. Line 46. Clarify what is meant by 'upstream', which is confusing in the context of waterborne infection. Line 51. Change 'was present with' to 'was identified as a result of'. Line 60. Omit 'as'. Line 60. Omit 'of'. Line 63. Omit 'filtered'. Lines 69-71.  This sentence needs rewriting. Lines 71-72. This sentence needs rewriting. Lines 72-73. Change to '...isolate from Farm B was the same ST (ST392) as had been found in the infected pig and the water samples'. Line 77. Change 'with' to 'of'. Figure 1. Caption needs some revision. Lines 96-98.  This sentence needs rewriting. Line 104. Change to 'Sprague and Neubauer reported in 2004 that...'. Line 110. Rewrite the sentence beginning 'Per a report from Malaysia...'. Line 121. Omit 'region' and change 'cautious' to 'aware'. Line 123. Change 'chemically clean' to 'disinfect'. Line 124. Surely it either is or is not part of the animal disease control program?

Author Response

Reviewer#2

Comments and Suggestions for Authors

This paper describes investigations into porcine melioidosis in sothern Thailand following the identification of an initial case in 2017.  A range of investigations (molecular epidemiology, environmental sampling, serology) was undertaken and a posible link to a water supply in one farm was made on the basis of these.

The manuscript would benefit from careful review by one of the authors whose first language is English.  Tenses in particular require attention and I have not attempted to list every instance where these need to be changed.  I have mentioned a few examples below where changes to the English would improve the paper.

Authors’ response: We appreciated the reviewer’s effort on help fixing most of the grammatical errors. The revised manuscript has been checked for proper grammar and tense usages.

My main comment is that Table 1 would benefit from the addition of some numbers (samples taken, samples positive, colonies tested by MLST etc.). 

Authors’ response: Table 1 has been modified per the reviewer’s suggestions.

It would also be useful if some more detailed information could be provided about the animals that were culture positive in 2018.  For example, did these have any links with Farms A and B, were the 2 ST 392 isolates from the same animal?

Authors’ response: Based on the information we recently received from the slaughterhouse, the two incidences on 06/05/18 were linked to Farm B. However, we could not confirm whether both specimens (lungs and liver) were from the same pig. This is because during the butchering process, all internal organs from multiple pigs were collected together into a big tray for inspection.

Since it is now 2 years after the initial case, it would also be useful to have some follow up on Farm B. Has it reopened, what happened about the water supply, have there been any more cases from there etc.?

Authors’ response: Farm B has been shut down since July 2018 and has no plan to reopen. B. pseudomallei was still found in the water supply when we did a follow up survey in May 2019 and the chlorination system has not been installed.

Minor suggestions to improve the paper:

Line 29. Change 'infection' to 'investigation'.

Authors’ response: This change has been made in the revised manuscript.

Line 33. Melioidosis is also endemic in other tropical areas.

Authors’ response: This change has been made in the revised manuscript.

Line 37. Change 'also is' to 'is also'.

Authors’ response: This change has been made in the revised manuscript.

Line 39. Reference 3 is not really relevant to this statement - Limmaturotsakul et al. Melioidosis in animals, Thailand, 2006-2010. Emerg Infect Dis, 2012 would be more appropriate.

Authors’ response: We noted that we used the irrelevant citation by error. This change has been made in the revised manuscript.

Line 40. Change 'zoonotic potentials' to 'the zoonotic potential'.

Authors’ response: This change has been made in the revised manuscript.

Lines 40-2. In fact there is very little hard evidence of zoonotic transmission of melioidosis and this section is purely hypothetical.  If these statements are to be included then they should be referenced.

Authors’ response: We agree with the reviewer that our statement is purely hypothetical. An additional reference has been provided.  

Line 46. Clarify what is meant by 'upstream', which is confusing in the context of waterborne infection.

Authors’ response: The word “upstream” has been removed to avoid causing confusion. 

Line 51. Change 'was present with' to 'was identified as a result of'.

Authors’ response: This change has been made in the revised manuscript.

Line 60. Omit 'as'.

Authors’ response: This change has been made in the revised manuscript.

Line 60. Omit 'of'.

Authors’ response: This change has been made in the revised manuscript.

Line 63. Omit 'filtered'.

Authors’ response: This change has been made in the revised manuscript.

Lines 69-71.  This sentence needs rewriting.

Authors’ response: This sentence has been rewritten in the revised manuscript.

Lines 71-72. This sentence needs rewriting.

Authors’ response: This sentence has been rewritten in the revised manuscript.

Lines 72-73. Change to '...isolate from Farm B was the same ST (ST392) as had been found in the infected pig and the water samples'.

Authors’ response: This sentence has been rewritten in the revised manuscript.

Line 77. Change 'with' to 'of'.

Authors’ response: This sentence has been rewritten in the revised manuscript.

Figure 1. Caption needs some revision.

Authors’ response: The cation has been revised.

Lines 96-98.  This sentence needs rewriting.

Authors’ response: This sentence has been rewritten in the revised manuscript.

Line 104. Change to 'Sprague and Neubauer reported in 2004 that...'.

Authors’ response: This change has been made in the revised manuscript.

Line 110. Rewrite the sentence beginning 'Per a report from Malaysia...'.

Authors’ response: This sentence has been rewritten in the revised manuscript.

Line 121. Omit 'region' and change 'cautious' to 'aware'.

Authors’ response: This change has been made in the revised manuscript.

Line 123. Change 'chemically clean' to 'disinfect'.

Authors’ response: This change has been made in the revised manuscript.

Line 124. Surely it either is or is not part of the animal disease control program?

Authors’ response: Yes, melioidosis in pigs is not currently a part of the animal disease control program in Thailand. We are working with the local animal health authority to develop a surveillance system of swine melioidosis through our Southern Thailand Melioidosis Initiative.

Round 2

Reviewer 2 Report

This paper has been improved but there are still numerous changes needed to the English to make it read better.  I did say in my previous report that I had only listed some of the changes needed and am disappointed to find that there is still so much editing required.

Lines 40-42.  Change to 'Choy and colleagues reported three possible zoonotic cases in Australia, including a butcher who directly handled infected meat or animals'. Line 42. Insert 'the' before 'zoonotic'. Line 45. Change to '...could represent a public health hazard'. Line 72-3. Change to 'identify the sequence type...'. Lines 73-5. Change to 'MLST analysis showed that all 36 B. pseudomallei isolates belonged to a single sequence type (ST392), which was the same as the isolates from the  infected pig...'. Lines 75-6. Change to 'In addition, one of the 17 soil samples collected from each farm grew B. pseudomallei.' Line 83. Change 'has shown' to 'showed'. Line 84-5. Change to 'all three of these STs have previously been reported from Thailand' Line 85. Change to 'to have been discovered'. Line 86-87. Change to 'This suggests that these porcine  infections were caused by local genotypes prevalent in southern Thailand'. Line 92-3. Change to 'and B. pseudomallei was isolated in pure culture on Ashdown's agar (b).' Line 95. Remove 'in' after 'soil'. Line 98-99. Change to 'where we had isolated B. pseudomallei from soil . A serological survey of antibodies to B. pseudomallei was...'. Lines 102-4. Change to '...that all piglets were serologically negative with titers ≤1:20, while most sows had titers ≤ 1:80, except a single sow with a titer of 1:320'. Line 108. Change 'due to' to 'as'. Line 120. Omit 'the'. Line 134. Omit 'infections'. Line 136-7. Omit 'To the best of our knowledge'.

Author Response

Authors’ responses to the reviewer’s comments:

Comments and Suggestions for Authors

This paper has been improved but there are still numerous changes needed to the English to make it read better.  I did say in my previous report that I had only listed some of the changes needed and am disappointed to find that there is still so much editing required.

Authors’ response: Thank you very much for the reviewer’s editorial comments. We very much appreciated your efforts.

Lines 40-42.  Change to 'Choy and colleagues reported three possible zoonotic cases in Australia, including a butcher who directly handled infected meat or animals'.

Authors’ response: This change has been made in the revised manuscript.

Line 42. Insert 'the' before 'zoonotic'.

Authors’ response: This change has been made in the revised manuscript.

Line 45. Change to '...could represent a public health hazard'.

Authors’ response: This change has been made in the revised manuscript.

Line 72-3. Change to 'identify the sequence type...'.

Authors’ response: This change has been made in the revised manuscript.

Lines 73-5. Change to 'MLST analysis showed that all 36 B. pseudomallei isolates belonged to a single sequence type (ST392), which was the same as the isolates from the  infected pig...'. Authors’ response: This change has been made in the revised manuscript.

Lines 75-6. Change to 'In addition, one of the 17 soil samples collected from each farm grew B. pseudomallei.'

Authors’ response: This change has been made in the revised manuscript.

Line 83. Change 'has shown' to 'showed'.

Authors’ response: This change has been made in the revised manuscript.

Line 84-5. Change to 'all three of these STs have previously been reported from Thailand'

Authors’ response: This change has been made in the revised manuscript.

Line 85. Change to 'to have been discovered'.

Authors’ response: This change has been made in the revised manuscript.

Line 86-87. Change to 'This suggests that these porcine  infections were caused by local genotypes prevalent in southern Thailand'.

Authors’ response: This change has been made in the revised manuscript.

Line 92-3. Change to 'and B. pseudomallei was isolated in pure culture on Ashdown's agar (b).'

Authors’ response: This change has been made in the revised manuscript.

Line 95. Remove 'in' after 'soil'.

Authors’ response: This change has been made in the revised manuscript.

Line 98-99. Change to 'where we had isolated B. pseudomallei from soil . A serological survey of antibodies to B. pseudomallei was...'.

Authors’ response: This change has been made in the revised manuscript.

Lines 102-4. Change to '...that all piglets were serologically negative with titers ≤1:20, while most sows had titers ≤ 1:80, except a single sow with a titer of 1:320'.

Authors’ response: This change has been made in the revised manuscript.

Line 108. Change 'due to' to 'as'.

Authors’ response: This change has been made in the revised manuscript.

Line 120. Omit 'the'.

Authors’ response: This change has been made in the revised manuscript.

Line 134. Omit 'infections'.

Authors’ response: This change has been made in the revised manuscript.

Line 136-7. Omit 'To the best of our knowledge'.

Authors’ response: This change has been made in the revised manuscript.